# 3D Genome Plasticity in Normal and Diseased Neurodevelopment

**DOI:** 10.3390/genes13111999

**Published:** 2022-11-01

**Authors:** Amara Plaza-Jennings, Aditi Valada, Schahram Akbarian

**Affiliations:** 1Graduate School of Biomedical Sciences, Medical Scientist Training Program, Icahn School of Medicine at Mount Sinai, New York, NY 10029, USA; 2Department of Psychiatry, Icahn School of Medicine at Mount Sinai, New York, NY 10029, USA; 3Department of Neuroscience, Icahn School of Medicine at Mount Sinai, New York, NY 10029, USA; 4Department of Genetics and Genomic Sciences, Icahn School of Medicine at Mount Sinai, New York, NY 10029, USA

**Keywords:** Hi-C, chromosomal conformations, 3D genome, cis-regulatory domain, brain, neurodevelopment, 4D nucleome, non-coding DNA, neuropsychiatric disorder

## Abstract

Non-random spatial organization of the chromosomal material inside the nuclei of brain cells emerges as an important regulatory layer of genome organization and function in health and disease. Here, we discuss how integrative approaches assessing chromatin in context of the 3D genome is providing new insights into normal and diseased neurodevelopment. Studies in primate (incl. human) and rodent brain have confirmed that chromosomal organization in neurons and glia undergoes highly dynamic changes during pre- and early postnatal development, with potential for plasticity across a much wider age window. For example, neuronal 3D genomes from juvenile and adult cerebral cortex and hippocampus undergo chromosomal conformation changes at hundreds of loci in the context of learning and environmental enrichment, viral infection, and neuroinflammation. Furthermore, locus-specific structural DNA variations, such as micro-deletions, duplications, repeat expansions, and retroelement insertions carry the potential to disrupt the broader epigenomic and transcriptional landscape far beyond the boundaries of the site-specific variation, highlighting the critical importance of long-range intra- and inter-chromosomal contacts for neuronal and glial function.

## 1. Introduction

The human brain is comprised of an estimated 170 billion neurons and glia, each of which contains a vast amount of genomic information stored in >6 gigabases of diploid genome. According to recent estimates, up to 80% of the human genome sequence undergoes low level transcription [1], and 7.9% of the genome sequence is directly associated with one of the 926,535 candidate cis-regulatory elements (CREs) [2], mostly comprised of ~10^3^ base pair stretches of non-coding DNA implicated in the regulation of gene expression. To date, many types of studies assess each gene or regulatory unit including promoters and enhancers as distinct entities, which when surveyed on a genome-wide level, are then further analyzed by some type of functional pathway or gene ontology-based analyses. Such type of approach has been extremely valuable for constructing cell- and circuit-specific ‘registries’ from a small number of reference brains [3] and for aligning heritability maps to non-coding regulatory DNA and specific cell types. For example, excitatory and inhibitory forebrain neurons carry a disproportionate share of heritability risk in schizophrenia and related disease [4,5,6], and likewise, microglial, vascular and perivascular genomes are overrepresented in the genetic risk map for Alzheimer’s neurodegeneration [7,8]. However, because the chromosomal material is spatially organized in a highly non-random manner inside neuronal, glial, and other non-neuronal nuclei, the brain’s transcriptomic and epigenomic landscapes are increasingly assessed in the context of this ‘3D genome’ or ‘spatial genome’. To this end, the various types of chromosomal conformations, mostly defined as intra- and inter-chromosomal contacts are commonly assessed by genome-scale ‘Hi-C’ DNA-DNA proximity assays (Figure 1A,B). These Hi-C based 3D genome maps, when superimposed and combined with epigenomic and transcriptomic maps, then allow for highly integrative studies that analyze the chromosomal organization in the context of the underlying epigenomic landscape. The epigenomic landscape includes many different modifications that range widely in terms of their scale. On the smaller end, there are single nucleotide-level modifications, including cytosine methylation, and 10^2^–10^3^ bp wide nucleosomal arrays marked by specific histone modifications associated with regulatory elements, such as promoters and enhancers. Larger scale epigenomic modifications include 10^6^ bp scaling chromosomal domains typically encompassing multiple transcriptional units, and ultimately the 6.4 × 10^9^ bp 3D nucleome, also referred to as ‘4D nucleome’ when assessed in context of dynamic changes during development and differentiation [9,10] (Figure 1C). While these integrative genomics concepts now broadly invade all areas of metazoan biology including the neurosciences, here in this review we provide a concise overview on some of the recent developments in the field of 3D genomics as it pertains to normal and diseased brain development and aging.

### 1.1. Spatial Chromosomal Organization and Neuroplasticity in Development

The structural organization of chromosomes, beyond the ‘beads-on-string’ elementary composition of chromatin fibers as arrays of nucleosomes interconnected by linker DNA, is broadly guided by two independent folding principles, namely phase separation (PS)-driven molecular forces and actively regulated loop extrusion mechanisms. An illustrative example for PS is provided by the intra- and inter-chromosomal polymer-polymer ‘lattice’ of heterochromatin-associated protein 1 (HP1) bound to nucleosomes tagged with a specific type of repressive histone mark [13]. PS has been ascribed to the formation and maintenance of various chromatin bodies, such as the nucleoli for ribosomal biogenesis, the nuclear speckles as splicing factories, and various types of heterochromatic clusters assembled by the aforementioned HP1 bridges and other types of repressor proteins, including the Polycomb complex [14,15]. On a genome-wide scale, PS mechanisms are likely to operate on 10–25 kb wide fragments predicted to underly the spatial segregation of active and inactive chromatin territories [16], which appear on chromosomal contact matrices constructed from Hi-C assays as a plaid pattern corresponding to active ‘A’ and silent ‘B’ compartments. It is generally accepted that these principles of 3D genome organization broadly apply to all somatic tissues in metazoans, including the developing nervous system. Consequently, genomic loci undergoing developmentally regulated switches between the A and B compartments are associated with corresponding changes in gene expression. 

In contrast, ATP-dependent loop extrusion mechanisms, mediated by cohesion, condensin, and various other chromatin-and DNA-binding proteins, such as CTCF, furnish kilo- to megabase-scaling chromosomal loop formations and the ~10^5^ base pair-scaling topologically associating domains (TADs) with their characteristic ‘folded-upon-self’ appearance in Hi-C chromosomal contact matrices [17,18]. Enhancers and genes sharing the same TAD interact more frequently with each other than with regions located elsewhere in the genome [19] and show much stronger correlations in epigenetic marks of chromatin activity [20]. Loops and TADs are generally important for furnishing proper interactions between promoters and distal CREs and likely for reducing the level of cryptic exogenic and intergenic transcription at the site of active enhancers [17,18]. 

The aforementioned principles are likely to apply to all somatic tissues including brain. Thus, developmentally regulated A/B compartment switching in the differentiating brain closely tracks the corresponding gene expression changes. This association has been broadly confirmed by recent studies mapping the spatial genome and the transcriptome in proliferative as compared to the differentiated layers of fetal human cerebral cortex [21], in mouse models of the neural precursor-to-neuron transition [22], and even in various neuronal subtypes of the developing retina [23]. These brain-specific developmentally regulated compartment switches are in excellent agreement with the notion that A/B compartmentalization closely tracks the epigenetic state and activity status of local chromatin fibers. 

In addition to compartment switching at the site of developmentally regulated genes, the differentiation of neural precursors into neurons is accompanied by widespread remodeling of chromosomal loop formations, specifically a shift away from shorter range loopings to more long-range (>100 kb) loops [24], and rearrangement of 3D chromatin structures at many neuron-specific genes [25]. As a result, there is significant widening of the average neuronal TAD length by at least 10% on a genome-wide scale [24]. This maturation-dependent right shift of TAD length and loop size in neurons resonates well with the fact that expression levels of longer gene transcripts are significantly higher in neural compared to non-neural tissues [26]. Also consistent with this observation, proper expression levels of many activity-regulated neuronal genes, particularly those with wider genomic distances between the two loop anchors, is heavily dependent on cohesin occupancy at loop anchors [27].

Furthermore, 3D genome organization in the young brain shows a surprising degree of environmental sensitivity. This is best illustrated in studies comparing chromosomal conformations in cerebral cortex and hippocampus of adolescent and young adult mice reared in enriched cages compared to standard cage housing. Housing in an enriched cage, just like challenging animals with a cognitive task or drug-induced synaptic activation, triggered genome-scale changes in transcription-associated chromatin accessibility and histone acetylation together with widespread remodeling of chromosomal conformations affecting promoter-enhancer loopings and compartment organization across hundreds of loci [28,29,30]. 

There is also increasing evidence that, beyond this initial critical period in the young developing brain, chromosomal conformations remain highly plastic during a wide window of brain development and aging. One intriguing example for spatial genome plasticity recently emerged from a chromosomal conformation study on ventral hippocampal neurons of the adult female brain in which they compared 3D genome plasticity across the estrous cycle, and also to the male brain. The study discovered evidence for a hormone-cycle driven decondensation of many X chromosomal domains, together with highly dynamic changes in promoter-enhancer loopings at hundreds of autosomal loci occupied by estrogen-associated transcription factors and nuclear hormone receptors [31]. Specifically, serotonin signaling and anxiety emerged as top functional pathways for these hormone cycle-sensitive chromosomal conformations [31]. 

### 1.2. Spatial Genome Organization and Brain Evolution

Many studies have also compared 3D genome structure across the evolutionary spectrum. Notably, chromosomal conformation mappings across multiple mammalian lineages have confirmed that TAD organization is robustly conserved across syntenic blocks of genome, an effect ascribed to the preservation of CTCF binding sites at TAD boundaries, even in genomes evolutionarily separated by tens of millions of years [32]. This general conservation of TAD boundaries across different species is much more robust for some of the larger megabase-scaling TADs than for some of the smaller-sized TADs. This is thought to reflect pressure from negative selection, because intra-TAD breaks, in contrast to breaks at the boundaries, are much more likely to disrupt the regulatory architecture of the affected TAD, and thereby much more likely to negatively affect viability of the mutant organism. However, comparative 3D genome mapping across mammalian tissues [32] and in the proliferative and neuronal layers of the developing prenatal cortex of primates and rodents have confirmed that the majority of species-divergent chromosomal conformation changes are linked to intra-TAD changes in CTCF binding and the emergence of smaller de novo TADs [33]. Interestingly, the emergence of human-specific chromosomal loopings is strongly enriched at the site of neurodevelopmental risk genes, and often linked to de novo enhancer and other cis-regulatory non-coding sequences [33,34], with many of the genes targeted by de novo loopings preferentially expressed in the subplate, a transient neuronal structure beneath the immature cerebral cortex critically important for cortical and thalamic circuit formation [33]. 

Many of these human-specific chromosomal conformation changes in the brain are linked to phylogenetically young retroelements which propagate throughout the genome via cut-and-paste mechanisms involving RNA intermediates. These retroelements include various Alu, LINE and HERV subtypes [33,35,36] that contain sequences epigenomically decorated with active histone marks and CTCF [37]. Similar mechanisms may shape neuronal and glial 3D genomes in the rodent lineage. In support of this, a recent study [38] reported that cell-type specific adaptations of chromosomal megadomains shape the 3D genome in mouse lines, particularly those with high genomic densities of phylogenetically young murine Endogenous Retrovirus (ERV) retroelements. They observed that locus-specific densities of ERV elements, which underwent a dramatic expansion in *Mus musculus*-derived inbred lines as compared to the wild-derived *Mus Spretus* inbred strain that diverged from *Mus musculus* 1.5 million years ago [39], are associated with much higher inter- and intra-chromosomal contact frequencies in neuronal chromatin [38]. This effect was absent in the surrounding non-neuronal (incl. glial) cells, pointing to important genome-scale differences in the chromosomal contact map of mature neurons even between closely related mouse species, and could be a reflection of neurotoxic and neuroinflammatory potential of ERV retroelements when unsilenced [38]. 

### 1.3. Neurological Disease Associated with Disrupted Organization of the 3D Genome

Disease-associated changes in the brain’s 3D genomes can be divided into three broad categories. The first category refers to monogenic neurodevelopmental disorders with structural mutations in chromatin architectural proteins, that typically result in loss of function for an essential component of the chromosomal scaffolding machinery. A classical, and frequently cited example is Cornelia de Lange Syndrome (CdLS), a neurodevelopmental disorder defined by intellectual disability and autism, often in the absence of gross morphological changes in the brain. The large majority of CdLS patients are affected by structural variation in cohesin complex genes or the cohesion loading factor NIBPL [40]. Importantly, neurons from CdLS postmortem brain, and from an experimental model for neuronal cohesin deficiency, show preferential gene expression changes at many neuronal genes regulated by distal CREs. These include a group of protocadherin cell adhesion molecules, all of which appear to require long-range promoter-enhancer loopings to reach proper levels of expression [41]. Other well-known examples include a spectrum of mutations in the CTCF coding sequence that are associated with a broad range of intellectual and behavioral disabilities, microcephaly, or multi-organ involvement [42]. Another interesting example is SATB2, a chromatin protein that assembles into dimers and tetramers and functions as loop organizer in neuronal nuclei, while also tethering chromosomal fibers to the inner nuclear membrane [43]. In addition to these selected examples, given that regulatory mechanisms in the nucleus typically encompass multiple layers of the epigenome, we would predict that mutations in a wide range of chromatin proteins associated with neurodevelopmental processes would demonstrate alterations in the brain’s chromosomal conformations. 

The second category of 3D genomic disruption is comprised of localized structural DNA mutations that cause the disruption of domain boundaries and the reconfiguration of intra- and inter-chromosomal contacts of the affected domain. This category includes copy number variant (CNV) syndromes associated with neuropsychiatric disease, including the 22q11 and 1q21.1 microdeletion syndromes which, remarkably, include severe alterations and disruptions of chromosomal connectomes that extend beyond the deletion site and its immediate flanking regions. Indeed, alterations in the chromosomal contact map emerge across the entire length of the affected chromosome as well as alterations to *trans*, or inter-chromosomal, connectivity [44]. 

This second category of 3D disease in the brain also include numerous examples of disrupted TAD domain boundaries. Notably in the primate lineage, genomic rearrangements at TAD level disproportionally occur at TAD boundaries, and include neurodevelopmental risk genes, such as *LYPD6*, which is involved in WNT/β catenin signaling [45]. This apparent increased vulnerability of TAD domain boundaries for DNA structural variation may be linked to the fact that boundary sequences frequently function as origins of DNA replication in or around the S-phase of the cell cycle [46,47]. In any case, it is remarkable that 22 out of 27 neurological and medical conditions associated with a locus-specific abnormal expansion of short tandem repeat (STR) (also known as microsatellites) defined by CGG, CAG, GCG and CTG repeat units, have their disease-associated STR positioned at the site of a TAD boundary [48]. This list includes neurodevelopmental and neurodegenerative conditions, including Fragile X syndrome with abnormal repeat expansions at a TAD boundary upstream at the *FMR*1 gene, Friedreich’s ataxia at the FXN gene, Huntington’s disease at the HTT gene, a type of motor neuron disease at the C9ORF72 locus, and spinocerebellar ataxia type 1 at the site of the ATXN1 gene [48]. Importantly, cell lines from Fragile X patients show altered chromosomal conformations at the affected TAD boundary with abnormal CTCF peak profiles within 100 kb of the abnormally expanded STR (Figure 2A) [48]. These probably operate in concert with other types of maladaptive epigenomic changes at the repeat expansion site, such as DNA hypermethylation. Ultimately, abnormal regulation across multiple epigenomic layers then very likely contributes to abnormal expression and silencing of the transcript for the fragile X mental retardation protein (FMRP), a widely expressed RNA-binding protein essential for proper synaptic plasticity and architecture [49]. Even more remarkably, these localized alterations in 3D chromatin structure at the Fragile X locus result in excessive spreading of heterochromatin upstream of the disease-associated STR, with the abnormal heterochromatic spread engulfing two X-linked neuronal cell adhesion genes, *SLTIRK2* and *SLTIKR4* [50], in addition to various autosomal loci interconnected in *trans* with the Fragile X locus, altogether affecting more than 10 Mb of genome sequence [50]. Therefore, similarly to the microdeletion syndromes, chromosomal conformation changes at neurologically relevant short tandem repeat expansions are not limited to the affected site-specific locus, but also affect additional loci that are in spatial 3D proximity to the specific repeat expansion site. This principle is likely not limited to the Fragile X site, because similar types of alterations of local chromatin structures and TAD boundary strength were reported in the striatum of a mouse model for the abnormal CAG repeat expansion at the HTT Huntington’s disease locus, together with significant chromosomal conformation changes at additional loci important for neuronal function [51]. Guided by these concepts then, the perception of these repeat expansion disorders as ‘monogenic’ may have to revised to integrate genomic and functional analyses of the genes spatially interconnected with the FMR1 and HTT, and other neurologically relevant STR loci, in 3D.

The third category of genome-scale chromosomal disorganization is exemplified by a group of cognitive and psychiatric disorders with typical onset in early adulthood, including schizophrenia and depression, which are often viewed as neurodevelopmental in origin. These disorders show complex genetic etiologies, and according to monozygotic twin studies from large national registries, carry a heritable risk ranging as high as 80% with twin concordance rates around 33% [52]. The hypothesis that dysregulation of chromatin structure and function could be the critical mechanism of disease for many of the affected cases is strengthened by the recent discovery that regulators of nucleosomal histone modifications and active chromatin status rank as one of the top scoring biological pathways in genome-wide association studies (GWAS) of schizophrenia (SCZ) and related co-heritable traits [53]. In addition, a number of rare mutations in a subset of histone methyl- and acetyl-transferase enzymes operating at sites of neuronal gene expression are observed in psychiatric diseases with high penetrance [54,55]. Furthermore, when schizophrenia heritability maps, which are constructed from common polymorphisms identified by GWAS in population-wide cohorts, are aligned with brain-specific histone modification and other epigenomic and transcriptomic maps, strong links between heritability risk and brain-specific promoter and enhancer sequences emerge [56]. Chromosomal conformation mappings in reference brains have further refined these concepts and defined them from a 3D genome perspective. For example, it was reported that there is a high proportion of chromosomal contacts that physically link non-coding DNA harboring expression quantitative trait loci (eQTL) to their predicted target genes, thereby bypassing the DNA on the linear genome [21,57,58]. Interestingly, these loop-dependent, risk-associated regulatory non-coding DNA sequences have been frequently linked to fetal development and transcriptional regulation of cortical projection neurons, in addition to various other neuronal subtypes [4,5,6,11,59,60]. Furthermore, genomic loci conferring schizophrenia heritability show a disproportionate physical association with nuclear speckles [61], a finding that would be in line with the existence of widespread splicing defects commonly observed in schizophrenia postmortem brain [62]. Because nuclear speckles generally tend to be located in the nuclear interior, away from the periphery, the spatial positioning of disease-associated chromosomal domains overall is significantly tilted towards the nuclear interior. These findings, taken together, imply that speckle-bound chromosomal loci could make significant contributions to schizophrenia genetic etiology and molecular disease manifestation.

However, these studies mapped schizophrenia heritability loci onto the 3D genome of a few reference brains. Therefore, it is still unclear whether chromosomal organization is altered in the diseased brains. To this end, a recent study, generating 739 histone acetylation and methylation profiles from prefrontal cortex of schizophrenia, bipolar and control subjects, identified thousands of cis-regulatory domains (CRDs), with each CRD defined by the coordinated regulation of sequentially positioned acetyl- and methyl-histone peaks distributed across a ~10^5^ base pair range, firmly embedded within the Hi-C based TAD chromosomal domain landscape [63]. Notably, CRD boundaries frequently were marked by CTCF occupancy, a protein which among various other functions is a key organizer of chromosomal foldings and loopings, including insulation of domain boundaries [64]. These findings then point to a novel level of chromosomal organization in brain, with CRDs representing a functional module embedded within TADs. Notably, in the aforementioned schizophrenia postmortem study [63], up to 2000 CRDs showed altered acetylation levels in neuronal chromatin of schizophrenia and bipolar PFC, often in conjunction with abnormal gene expression in cis. Intriguingly, while hypoacetylated CRDs showed strong 10:1 over-representation of loci driving inhibitory interneuron function, hyper-acetylated CRDs were specifically aligned with the GWAS-defined schizophrenia risk map and excitatory (prefrontal projection neuron) signaling (Figure 2B) [63]. A subset of hyperacetylated CRDs showed evidence for functionally and structurally defined interconnectedness within the 3D nucleome of PFC neurons, based on correlational clusterings across individuals using Euclidean distance-based spatial proximity estimates modeled from Hi-C chromosomal conformation mappings [63]. Interestingly, these disease-sensitive CRDs in neurons from the adult PFC of subjects with schizophrenia harbor a number of sequences that show high levels of acetylation during normal prenatal development, which would suggest that at least some of the CRDs hyperacetylated in the adult diseased brains represent vestiges of early occurring neurodevelopmental disruptions [63]. Thus, the diseased fetal histone peaks could serve as seed points for altered histone acetylation that ultimately could affect the entire extent of the CRD, and even spill into other CRDs that colocalize in 3D space in the nucleus. 

From this perspective, chromosomal domains could be viewed as information storage units that maintain, in the adult brain, epigenetic scars of neurodevelopmental defects that occurred many years prior to the onset of disease. There is evidence that some of the other regulatory layers of the epigenome also harbor clues for a neurodevelopmental etiology of disease in schizophrenia. For example, DNA methylation profiling for 456,000 autosomal CpG dinucleotides in more than 520 prefrontal cortex (PFC) specimens collected from 14th week of gestation to 80 years of age, including 191 adult specimens from subjects diagnosed with schizophrenia, revealed that while the DNA methylation alteration in the affected brains only affected a minute (<0.5%) fraction of the overall CpG pool, an overwhelming majority of these disease-sensitive CpGs showed evidence for dynamic methylation changes during the transition from the pre- to the postnatal period of normal development [65]. Interestingly, independent studies reported a subtle association of developmentally regulated methyl-CpGs with the GWAS-based genetic risk map of schizophrenia. Therefore, both of these approaches—DNA methylation mappings comparing case and control brains, and comparison of the GWAS heritability maps with the developmentally regulated PFC DNA methylome—point to a potential disease-relevant role of regulatory non-coding sequences in the developing cortex. 

### 1.4. 3D Genome Organization in Context of Infection and Neuroinflammation

Intriguingly, a number of viral infections have recently emerged as examples of 3D genome dysregulation in the nervous system, including the SARS-CoV-2 infected sensory olfactory epithelium [66]. This virus, which is responsible for COVID-19 related illnesses, elicits, in contrasts to other types of upper respiratory tract infections, olfactory deficits, including severe anosmia, that are not explained by conductive interference [67]. Instead, SARS-CoV-2 infection confers, in a non cell-autonomous manner, a robust downregulation of odorant receptor gene expression in the olfactory sensory neurons of the infected host. This is associated with severe disruption of promoter-enhancer loopings and other intra- and inter-chromosomal contacts that bundle together the euchromatic enhancer islands regulating transcriptionally active odorant receptor allele in an healthy olfactory sensory neuron [66,68,69]. 

SARS-CoV-2 is likely not the only virus to disrupt 3D genome organization in susceptible brain cells. For example, Human Immunodeficiency Virus (HIV), an exogenous Long-terminal-repeat (LTR) type of retrovirus that stably inserts itself into the host’s nuclear genome, infects cells in brain, primarily microglia and perivascular macrophages. Ultimately, HIV infection can cause a wide range of neurological conditions that exist along the spectrum of HIV-associated neurocognitive disorders (HAND), which affect 20–50% of the 37 million people living with HIV (PLWH) [70,71,72]. Strikingly, microglia from brain tissue of individuals diagnosed with HIV encephalitis show a large scale reorganization of their chromosomal conformations, with altogether 192 megabases of microglial genome (exceeding the total length of human chromosome 5, to provide a dimensional perspective) affected. This is primarily driven by a shift of interferon-sensitive genes that are positioned within loci that convert into a more permissive open-chromatin-associated compartment status, as defined by significant changes in A/B compartment organization [73]. These microglia-specific 3D genome changes in the HIV infected human brain could be partially modeled by interferon stimulation of microglia in cell culture. Furthermore, many of the genomic loci that were newly mobilized into a more active compartment environment were at increased risk for being targeted for retroviral insertion, revealing a highly dynamic interrelationship of interferon-associated 3D genome and transcriptome remodeling with HIV integration in the brain [73]. 

## 2. Conclusions

Here, we review a number of recent discoveries in the field of 3D genomics and nucleomics, which highlight the delicate regulation of spatial genome organization including chromosomal folding and intranuclear topographies, during normal brain development and disease. The 3D genome organization in neurons and glial cells emerges as a highly cell type-specific layer of epigenomic organization, broadly relevant for the cell’s transcriptional programs via the furnishing of promoter-enhancer loopings or silencing of retrotransposons and cryptic transcription from intergenic sequences. Deeper exploration of the brains 3D genomes will also be highly relevant for the molecular pathology of a number of neurodevelopmental and neurological conditions. This is because it is increasingly recognized that alterations in genome organization and function due to localized structural variations of the genomic DNA, including micro-deletions and -duplications or abnormal repeat expansions, are not limited to the site of mutation and its immediate surroundings but could impact, via intra- and inter-chromosomal conformations, gene expression programs at additional loci. Similarly, as exemplified by SARS-CoV-2 exposed olfactory neuroepithelium and HIV exposed microglia, infection and inflammation is capable of triggering, in susceptible brain cells, a significant reorganization of chromosomal conformations.

## Figures and Tables

**Figure 1 genes-13-01999-f001:**
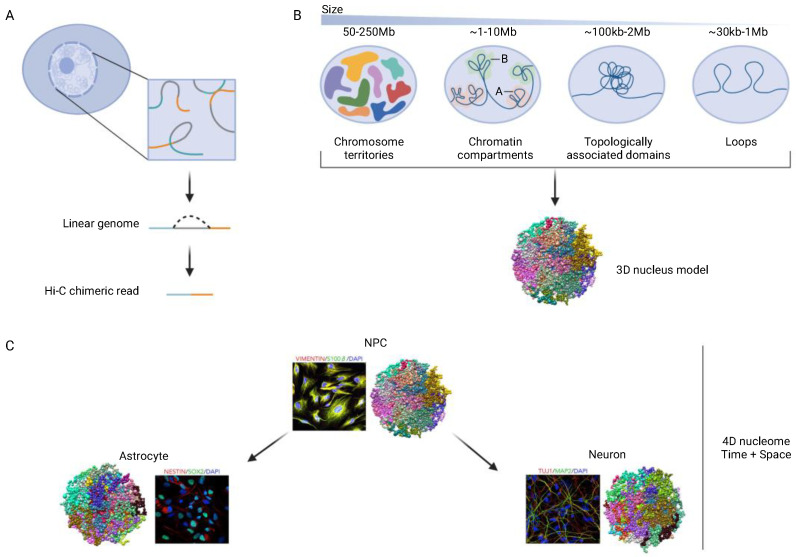
Hi-C Chromosomal contact (DNA-DNA proximity) mapping. (**A**) Basic principle. DNA-DNA proximity mappings typically use restriction digest followed by proximity religation techniques to capture chromosomal contacts. (**B**) basic building blocks of the 3D genome in the kilo- and megabase range. (**C**) chrom3D in silico model of topologically associated domains (TADs) [11,12], 1 ball = 1 TAD, TADs from same chromosome share the same color. The 4D nucleome is a concept for comprehensive, integrative analysis of the nucleus across time, including changes during differentiation.

**Figure 2 genes-13-01999-f002:**
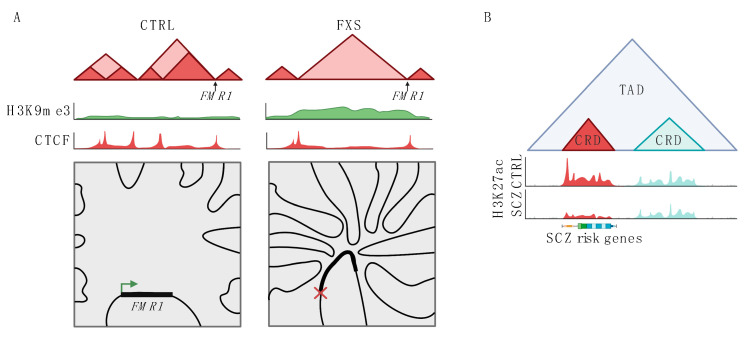
Changes to 3D genome structure in neuropsychiatric disease. (**A**) Schematic representation of changes to the *FMR1* locus in fragile X syndrome (FXS). In FXS, normal TAD structure (shown on the left, CTRL) is disrupted. This change to TAD structure is associated with increased H3K9me3 levels and deceased CTCF binding across the disrupted region. In FXS, the *FMR1* locus also has increased trans interactions with multiple other chromosomes and decreased expression. (**B**) Schematic showing changes to CRDs in schizophrenia (SCZ). Two CRDs are shown within one TAD, one of the CRDs (shown in red) has lower H3K27ac levels in SCZ as compared to health controls (CTRL). This CRD contains genes associated with SCZ risk in GWAS studies. The other CRD is unchanged in SCZ (shown in blue).

## Data Availability

Not applicable.

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
