# Peer review of "3D Genome Plasticity in Normal and Diseased Neurodevelopment"

_genes, 2022, doi:10.3390/genes13111999_

Round 1

Reviewer 1 Report

In this review, Plaza-Jennings and colleagues present and discuss several cases in which 3D Genomics contributed to propose the possible origin of brain cell diseases. I commend the authors for offering valuable and informative examples in which variations of the genome organization are related to different gene regulations and, eventually, to medical conditions in brain cells.

Minor points

1 – The authors pleasantly recalled the works that, applying 3D genomics methods, elucidate the relationship between genome architecture and genome function in brain cells. However, I encourage them to comment on a paper by Winick-Ng and colleagues (Nature 599, 684–691 (2021)) that characterizes the genome architecture in mouse brain cells, highlighting high specificity between chromatin organization and gene function in different cell types.

2 – In Lines 308-309, please consider revising the sentence “However, while these studies firmly link the genetic risk architecture of schizophrenia and related disorders to the brain’s 3D genome, it remained unclear whether chromosomal organization is altered in the diseased brains”. The word “architecture” is not connected to rest of the sentence.

3 – In Lines 369-370, please consider correcting this part of a sentence “but could impact, via intra- and inter-chromosomal conformations, potentially impact gene expression programs at 370 additional loci”.

Author Response

1 – The authors pleasantly recalled the works that, applying 3D genomics methods, elucidate the relationship between genome architecture and genome function in brain cells. However, I encourage them to comment on a paper by Winick-Ng and colleagues (Nature 599, 684–691 (2021)) that characterizes the genome architecture in mouse brain cells, highlighting high specificity between chromatin organization and gene function in different cell types.

Response: We appreciate this comment and now remark on (and cite) Winick-Ng et al. 2021) (line 120)

2 – In Lines 308-309, please consider revising the sentence “However, while these studies firmly link the genetic risk architecture of schizophrenia and related disorders to the brain’s 3D genome, it remained unclear whether chromosomal organization is altered in the diseased brains”. The word “architecture” is not connected to rest of the sentence.

Response: We appreciate this comment and now rephrased (line 313, 314) “However, these studies mapped schizophrenia heritability loci onto the 3D genome of a few reference brains. Therefore, it still remains unclear whether chromosomal organization is altered in the diseased brains.

3 – In Lines 369-370, please consider correcting this part of a sentence “but could impact, via intra- and inter-chromosomal conformations, potentially impact gene expression programs at 370 additional loci”.

Response:We appreciate this comment and have corrected this sentence (it is line 413 in the revised version)

Reviewer 2 Report

The authors presents a review on the 3D genome plasticity in both normal and pathological development. The review is well written and list recent discoveries about how 3D genome topology can affect neurological development.

I have only suggestions to the authors which would be related to dedicate in the section relative to the transposable elements (and their role in 3D genome shaping) the space additional references such as:

https://pubmed.ncbi.nlm.nih.gov/33922141/

https://pubmed.ncbi.nlm.nih.gov/33659801/

As brain and related neurological districts can be affected by viral infection I would like to also see a small section of the review talking about very recent findings (https://pubmed.ncbi.nlm.nih.gov/35180380/) relating to the effects of viruses such as Sars-CoV2 in causing neurological and cognitive defects as the least understood symptoms of COVID-19 patients. This work finds that this virus alters the nuclear organization of neurological anatomical districts important for the onset of anosmia. I think it would be a nice addition to the review.

Author Response

I have only suggestions to the authors which would be related to dedicate in the section relative to the transposable elements (and their role in 3D genome shaping) the space additional references such as:

https://pubmed.ncbi.nlm.nih.gov/33922141/

https://pubmed.ncbi.nlm.nih.gov/33659801/

Response: We appreciate this comment and now have added both citations to the review (line 175)

As brain and related neurological districts can be affected by viral infection I would like to also see a small section of the review talking about very recent findings (https://pubmed.ncbi.nlm.nih.gov/35180380/) relating to the effects of viruses such as Sars-CoV2 in causing neurological and cognitive defects as the least understood symptoms of COVID-19 patients. This work finds that this virus alters the nuclear organization of neurological anatomical districts important for the onset of anosmia. I think it would be a nice addition to the review.

Response: We thank the Reviewer for this very constructive comment and in response, have included a new chapter (lines 364 to 398) discussing the impact of viral infection on 3D genome organization in the nervous system, and we cite the above Sars-CoV2 reference as requested by the Reviewer.